# Heparanase Inhibition Prevents Liver Steatosis in E_0_ Mice

**DOI:** 10.3390/jcm11061672

**Published:** 2022-03-17

**Authors:** Safa Kinaneh, Walaa Hijaze, Lana Mansour-Wattad, Rawan Hammoud, Hisam Zaidani, Aviva Kabala, Shadi Hamoud

**Affiliations:** 1Department of Physiology, Rappaport Faculty of Medicine, Technion—Israel Institute of Technology, Haifa 3200003, Israel; safakinaneh@gmail.com (S.K.); avivak@technion.ac.il (A.K.); 2Department of Emergency Medicine, Rambam Health Care Campus, Haifa 3109601, Israel; wal.has@hotmail.com (W.H.); h_zaidani@rambam.health.gov.il (H.Z.); 3Department of Internal Medicine E, Rambam Health Care Campus, Haifa 3109601, Israel; lana.640@hotmail.com; 4Faculty of Biotechnology, Hadassah Academic College, Jerusalem 9101001, Israel; rawan-h-96@outlook.co.il; 5Lipid Research Laboratory, Rappaport Faculty of Medicine, Technion—Israel Institute of Technology, Haifa 3200003, Israel

**Keywords:** heparanase, PG545, SST0001, liver steatosis, E_0_ mice

## Abstract

Background: Non-alcoholic fatty liver disease affects up to 30% of adults in the USA, and is associated with a higher incidence of chronic liver morbidity and mortality. Several molecular pathways are involved in the pathology of liver steatosis, including lipid uptake, lipogenesis, lipolysis, and beta-oxidation. The enzyme heparanase has been implicated in liver steatosis. Herein, we investigated the effect of heparanase inhibition on liver steatosis in E_0_ mice. Methods: In vivo experiments: Male wild-type mice fed with either chow diet (*n* = 4) or high-fat diet (*n* = 6), and male E_0_ mice fed with chow diet (*n* = 8) or high-fat diet (*n* = 33) were included. Mice on a high-fat diet were treated for 12 weeks with PG545 at low dose (6.4 mg/kg/week, ip, *n* = 6) or high dose (13.3 mg/kg/week, ip, *n* = 7), SST0001 (1.2 mg/mouse/day, ip, *n* = 6), or normal saline (control, *n* = 14). Animals were sacrificed two days after inducing peritonitis. Serum was analyzed for biochemical parameters. Mouse peritoneal macrophages (MPMs) were harvested and analyzed for lipid content. Livers were harvested for histopathological analysis of steatosis, lipid content, and the expression of steatosis-related factors at the mRNA level. In vitro experiments: MPMs were isolated from untreated E_0_ mice aged 8–10 weeks and were cultured and treated with either PG545 or SST0001, both at 50 µg/mL for 24 h, followed by assessment of mRNA expression of steatosis related factors. Results: Heparanase inhibition significantly attenuated the development of liver steatosis, as was evident by liver histology and lipid content. Serum analysis indicated lowering of cholesterol and triglycerides levels in mice treated with heparanase inhibitors. In liver tissue, assessment of mRNA expression of key factors in lipid uptake, lipolysis, lipogenesis, and beta-oxidation exhibited significant downregulation following PG545 treatment and to a lesser extent when SST0001 was applied. However, in vitro treatment of MPMs with PG545, but not SST0001, resulted in increased lipid content in these cells, which is opposed to their effect on MPMs of treated mice. This may indicate distinct regulatory pathways in the system or isolated macrophages following heparanase inhibition. Conclusion: Heparanase inhibition significantly attenuates the development of liver steatosis by decreasing tissue lipid content and by affecting the mRNA expression of key lipid metabolism regulators.

## 1. Background

Non-alcoholic fatty liver disease (NAFLD), the most common form of fatty liver disease, affects up to 30% of adults in the USA. More than 25% of subjects with NAFLD are assumed to have non-alcoholic steatohepatitis (NASH), characterized by fat accumulation (steatosis) in hepatocytes and elevated serum transaminase levels with no other obvious etiology for liver injury. Definitive diagnosis of NASH is based on histological evidence of steatosis in hepatocytes, liver-cell injury and death, accumulation of inflammatory cells, and liver fibrosis [1,2,3,4]. It is estimated that 2% of American adults with NASH will develop cirrhosis during their lives [5], making NASH a leading cause for liver transplantation in the US [6,7,8]. In addition, NASH is a leading etiology for the rising incidence and prevalence of primary liver cancer [6], which is expected to reach an annual incidence of 1–2% among patients with NASH cirrhosis [7].

Several factors are involved in lipid uptake by the liver, including LDL receptor (LDLR), LDLR-related protein (LRP), and CD36; and in lipid metabolism, including ACAT2, CPT1a, HMGCoA reductase, pPARα, pPAR-γ, PCSK9, ACAT1 and DGAT1 [9]. 

The main cells involved in the pathogenesis of liver fibrosis are activated Kupffer cells and hepatic stellate cells (HSCs). Kupffer cells initiate inflammatory and fibrogenic responses by releasing cytokines, chemokines, and growth factors that augment inflammation, resulting in the activation of HSCs. Factors released by activated Kupffer cells include TNF-α, IL-6, IL-1β, and TGF-β. TGF-β induces fibrogenesis through activation of HSCs and reactive oxygen substances, leading to inflammation and liver damage [10]. In addition, both Kupffer cells and hepatocytes are involved in the secretion of fibroblast growth factor-2 (FGF-2), which is secreted following liver insult and stimulates hepatocyte regeneration and growth, as well as HSC proliferation and activation. FGF-2 causes excessive extracellular matrix (ECM) deposition by HSCs, thus inducing tissue perturbation and disruption [11,12]. Quiescent HSCs transform to fibrogenic α-SMA positive myofibroblasts, which profoundly alter the microenvironment by secreting excessive ECM proteins (collagens, fibronectin, and laminin). Although these processes aim to recover impaired organ function, prolonged injury causes dysregulation of the regeneration process and results in uncontrolled fibrogenesis, ECM accumulation, and disrupted organ architecture [13,14,15,16]. 

The repair of an injured liver requires autophagy, a process by which cells degrade their own components by forming autophagosomes and autolysosomes [17]. It occurs in all types of cells and serves as a homeostatic mechanism to recycle proteins and organelles. LC3 is a biochemical hallmark of autophagic flux in HSCs, and increased LC3 expression is associated with accelerated HSC autophagy and cell death, resulting in decreased fibrogenesis in the liver [18].

Knowing that triglycerides (TG) are the main lipid component that leads to steatosis and exhibit direct hepatotoxicity, it is presumed that NASH may be the common manifestation of diverse disease processes. Therefore, characterization of these processes is essential for understanding the pathogenic mechanisms underlying NASH development and progression [19]. In fact, hepatocyte injury and death are key features that differentiate NASH from isolated steatosis [4]. Injured hepatocytes release factors that promote the accumulation of immune cells that produce hepatotoxic substances resulting in further injury and inflammation [20,21], where it may induce stress responses and eventually cell death [22,23]. Thus, agents that inhibit the recruitment of inflammatory cells, block inflammatory signaling, and reduce oxidative stress (OS) may provide beneficial effects for NASH patients [19].

The enzyme heparan sulfate endoglycosidase heparanase (HPSE) is the only enzyme in mammals that degrades heparan sulfate (HS) chains of heparan sulfate proteoglycans (HSPG) in the ECM and basal membranes. In its intracellular role, HPSE participates in the turnover of membrane-associated HSPGs, while the secreted enzyme is involved in the remodeling and degradation of ECM [24,25]. Thus, in physiological conditions, HPSE is tightly regulated to prevent uncontrolled HS cleavage and adverse biological effects. It is well documented that HPSE activity is involved in glomerular basement membrane disassembly and proteinuria in several glomerulopathies [26,27] and in angiogenesis and tumor cell migration in cancer progression [28,29]. It was also shown that by regulating the bioavailability and activity of growth factors (FGF-2 and TGF-β), HPSE promotes tubular epithelial-to-mesenchymal transition (EMT) of proximal tubular cells and kidney fibrosis [30,31], and in streptozotocin-induced diabetic nephropathy model, HPSE-KO mice showed less interstitial fibrosis compared to untreated mice [32]. 

While the role of HPSE as a pro-cancerous agent has been well characterized in hepatocellular carcinoma (HCC) [33,34,35], its involvement in non-malignant chronic liver conditions is poorly understood, and controversial results were reported both from human tissues and animal models of liver fibrosis. Xiao et al. did not detect significant differences in HPSE mRNA and protein levels between normal and cirrhotic livers, compared to increased levels in HCC [34]. On the other hand, Ikegucli et al. documented a decreased HPSE mRNA level in HCC tissue compared to adjacent non-cancerous tissue, and HPSE expression in non-cancerous tissue even negatively correlated with fibrosis stage [36]. 

Apolipoprotein E deficient mice (E_0_/E^−/−^ mice) are an atherogenic animal model that develops accelerated atherosclerosis, especially when placed on a high-fat diet (HFD). The effect of heparanase inhibition by novel inhibitors is under extensive investigation in health and disease states. Of these inhibitors, PG545, a tetrasaccharide heparan sulfate mimetic molecule that inhibits heparanase activity, significantly decreased serum OS in the atherosclerosis animal model [37]. Notably, in our recent study with E_0_ mice placed on HFD, PG545 significantly reduced OS and both aortic wall thickness and aortic atherosclerotic plaque surface area, along with a significant reduction in liver steatosis and 95% decreased intracellular fat content in hepatocytes compared to control [38]. Likewise, SST0001, a 100% N-acetylated and glycol split non-anticoagulant heparin with potent anti-heparanase activity, had proved effective in reducing tumor progression and spread in multiple myeloma patients [39] and in Ewing’s sarcoma model [40], as well as in other human pediatric sarcoma models [41].

Therefore, we investigated herein the effect of heparanase inhibition on the development of liver steatosis and fibrosis in E_0_ mice placed on HFD, focusing on possible mechanisms by which heparanase inhibition exerts these effects. 

## 2. Methods

### 2.1. In Vivo Experiments: Animals

Male C57 Wild Type (WT) and ApoE^−/−^ (E_0_) mice aged 10–12 weeks were studied. Mice were housed in a pathogen-free environment under standard conditions at the Animal Care Facility of the Rappaport Faculty of Medicine. The study was conducted according to the National Institutes of Health guidelines and was approved by the Technion Ethics Committee for Experimentation in Animals—ethics number IL1090717. Mice were divided into seven treatment groups as follows:(1)ApoE^−/−^ fed with chow diet (CD), *n* = 8, (E_0_-CD);(2)ApoE^−/−^ fed with high-fat diet (HFD, TD.88137, 42% of calories from fat) and treated with saline, *n* = 14, (E_0_-HFD);(3)ApoE^−/−^ fed with HFD and treated with PG545 at low dose (0.2 mg/mouse/once weekly, 6.4 mg/kg), *n* = 6;(4)ApoE^−/−^ fed with HFD and treated with PG545 at high dose (0.4 mg/mouse/once weekly, 13.3 mg/kg), *n* = 7;(5)ApoE^−/−^ fed with HFD and treated with SST0001 (1.2 mg/mouse/daily), *n* = 6(6)WT fed with CD, *n* = 4;(7)WT fed with HFD, *n* = 6.

Treatment was applied for 12 weeks (Figure 1). Mice body weights and food intake were assessed weekly. Two days before sacrificing the mice, 4% thioglycollate was injected IP to induce peritonitis, which enabled us to isolate mouse peritoneal macrophages (MPMs), as described previously [37,38]. 

At the end of the experiment, mice were sacrificed. Briefly, following anesthesia with isoflurane, blood was withdrawn by direct puncture of the retro-orbital plexus using heparinized capillary tubes. Blood samples were centrifuged at 1600 rcf for 10 min, and serum was analyzed for biochemical parameters. The liver was collected in formalin or snap-frozen, and MPMs were isolated for subsequent assessment.

### 2.2. In Vitro Experiments

#### 2.2.1. MPM Isolation and Treatment

Three ml of 4% thioglycollate were injected, ip, to C57- ApoE^−/−^ mice (*n* = 3). Two days later, MPMs were isolated, and cells were combined and cultured in DMEM with 10% FBS for 24 h. Subsequently, cells were treated with either PG545 (50 µg/mL), SST0001 (50 µg/mL), or PBS for 24 h, then were harvested and analyzed for cholesterol and triglycerides mass assessment or for mRNA expression of steatosis-related factors (Figure 1). 

Histopathology: Mice livers were sliced into pieces. One piece was formalin-fixed and paraffin-embedded, and 5 µm-thick sections were stained with Hematoxylin–Eosin (H&E), Picrosirius red, and Masson’s Trichrome stainings. Another piece of the liver was snap-frozen in liquid nitrogen and further processed for Oil Red O staining. All stainings were performed according to standard protocols. The stained slides were scanned using 3D Histech Pannoramic MIDI (3D HISTECH Ltd., Budapest, Öv u. 3., Hungary). Representative images were viewed and captured using panoramic viewer software (3D HISTECH Ltd.). 

Immunofluorescence staining: Briefly, liver sections were rehydrated, followed by antigen retrieval using a proteinase K (Abcam# ab642201). Sections were blocked with 10% normal donkey serum (Cat# 017-000-121, Jackson ImmunoResearch, West Grove, PA, USA), then incubated with rabbit anti-heparanase antibody overnight or rat anti F4/80. Next, sections were incubated with Cy™3 donkey anti-rabbit secondary antibody (#711-165-152, Jackson ImmunoResearch, 1/100) or donkey anti-goat secondary antibody (#712-165-153, Jackson ImmunoResearch, 1/100). Liver sections stained with anti F4/80 were further stained with wheat germ agglutinin Alexa-fluor 488 (#W11261, Thermo Fisher Scientific, Waltham, MA, USA). Finally, all sections were mounted with DAPI Immunomount (Cat# 0100-20, SouthernBiotech, Birmingham, AL, USA) and were then visualized using Zeiss Axio observer inverted microscope system (Zeiss Axio, 37030 Göttingen, Germany).

#### 2.2.2. Serum Analysis 

Alanine aminotransferase (ALT), aspartate aminotransferase (AST), total cholesterol (TC), triglycerides (TG), high-density lipoprotein (HDL), and low-density lipoprotein (LDL) levels were determined using commercial kits (Siemens, Germany) with an auto-analyzer dedicated instrument (Dimension RXL, Siemens, Germany).

Oxidative stress (OS). For technical limitations, OS was performed only in the PG545 subgroups: 

PON1 arylesterase activity: Serum paraoxonase 1 (PON1) arylesterase activity was measured using phenylacetate as the substrate. Initial rates of hydrolysis were determined spectrophotometrically at 270 nm. The assay mixture included 5 µL of serum, 1.0 mmol/L phenylacetate, and 0.9 mmol/L CaCl_2_ in 20 mmol/L Tris HCl, pH 8.0. Non-enzymatic hydrolysis of phenylacetate was subtracted from the total rate of hydrolysis. The E_270_ for the reaction is 1310 mol/L^−1^·cm^−1^. One unit of arylesterase activity is equal to 1 micromole of phenylacetate hydrolyzed/min/mL.

Serum lipid peroxidation: Serum was incubated with 100 mM/L of the free radical generator 2,2′-Azobis(2-methylpropionamidine) dihydrochloride (AAPH) for 2 h at 37 °C, followed by thiobarbituric acid reactive substances (TBARS) analysis to assess peroxidation vulnerability.

TBARS: In order to determine the relative lipid peroxide content of mice serum, the TBARS method was used to measure malondialdehyde (MDA) products during an acid heating reaction. Briefly, the samples were mixed with a mixture of TBA and trichloroacetic acid in HCl to precipitate the protein. The reaction was performed at pH 2–3 at 90 °C for 20 min. The precipitate was pelleted by centrifugation at 4000× *g* at room temperature for 15 min. The absorption of the supernatants was read at a wavelength of 532 nm. The majority of TBARS are malondialdehydes; thus, the concentration of MDA in blood serum is expressed as nmol MDA/mL [42,43].

#### 2.2.3. Real-Time Polymerase Chain Reaction (qPCR)

Total RNA was extracted from frozen liver tissue from all treated animals using Tri-reagent (Life technologies #15596026, Frederick, MD, USA). RNA (5 μg) was reverse-transcribed into cDNA using the Maxima First Strand cDNA Synthesis Kit (Thermo Scientific #K1672, (EU), Vilnius, Lithuania). In order to assess a specific mRNA product, all the following components were mixed together in a total volume of 10 μL; cDNA template (15 ng), forward and reverse primers (0.2 μM), and SYBER green mixture (Cat# FP205-02, Tiagen, Beijing, China). The thermal conditions included a heat activation of the hotstart DNA polymerase at 95 °C for 15 min, followed by 40 cycles of denaturing at 95 °C for 10 s, annealing at 58 °C for 20 s, and extension at 72 °C for 30 s.

We used primers to detect key genes in lipid uptake, lipolysis, lipogenesis, beta-oxidation, inflammation, autophagy, and heparanase system. Forward and reverse primers sequences are listed in Table 1. Levels of a specific gene were quantified by the delta–delta CT method using RPL13a as a housekeeping gene.

#### 2.2.4. Cholesterol and Triglycerides Mass in MPM and Liver Tissue

Lipids from MPMs and liver tissue were extracted using Hexane: Isopropanol (3:2, *v*/*v*), and the hexane phase was evaporated under nitrogen. Cholesterol mass was determined using a CHOL kit (Roche Diagnostics GMbH, Mannheim, Germany), whereas the number of triglycerides was determined using a triglyceride determination kit (Sigma Aldrich, Burlington, MA, USA). Protein concentrations in liver tissue and MPM were measured using Lowry assay [44]. The results are expressed as μg cholesterol/mg cell protein or μg triglycerides/mg cell protein.

Statistical analysis: One-way analysis of variance (ANOVA) followed by the Tukey test was used for comparison of treatment values with baseline in each group or with corresponding values in the control group. All data analyses were conducted using GraphPad Prism version 5.03 (GraphPad Software, Inc., San Diego, CA 92037, USA). A value of *p* < 0.05 is considered statistically significant. Data are presented as mean ± SEM. 

## 3. Results 

Effect of heparanase inhibition on liver steatosis. PG545 significantly decreased liver fat content as compared to the control group in a dose-dependent manner. Moreover, SST0001 significantly attenuated the development of liver steatosis (Figure 2).

In the SST0001 treated group, large granuloma-like structures in the liver were observed, stained by Alexa flour staining, mostly demarcating HSC/activated KCs with increased lipid uptake (Figure 3).

In MPMs, treatment with either heparanase inhibitors significantly decreased both TC and TG content in the cells compared to E_0_-HFD control (Figure 4G,H).

Fat content in the treated groups was comparable to the E_0_-CD group.

In the liver tissue, heparanase inhibition resulted in significantly lower content of TC, LDL, HDL, and TG. Gene expression assays revealed that heparanase inhibition significantly decreased mRNA levels of LDLR, LRP, and CD36 (Figure 5A–C). Similarly, heparanase inhibition significantly reduced mRNA levels of DGAT1, MTTP, ApoB, and HMGCoA reductase (Figure 5E–H). There was no difference in mRNA expression of lipoprotein lipase (LPL).

In vitro studies on MPMs showed that heparanase inhibition was associated with increased cholesterol uptake by MPMs, and decreased IL6 and TNF-α to various extents (PG545 was more potent than SST0001, Figure 6). 

Inflammatory and oxidative reactions in the liver. The effect of PG545 and SST0001 on lipid oxidation was assessed by measuring levels of mRNA expression of CPT1a, pPAR1α, and pPAR-Ɣ. Heparanase inhibition resulted in significant decrease in CPT1a (as compared to HFD group, *p* < 0.01 for PG545 low dose, *p* < 0.001 for PG545 high dose, and *p* < 0.01 for STT0001). There was an increase in pPARc1a and a decrease in pPAR-Ɣ mRNA to variable extents (Figure 7A,C,D). Heparanase inhibition resulted in decreased mRNA expression of TLR4, IL6, and TNF-α, all significant for PG545 and non-significant for SST0001. TNF-α mRNA significantly decreased in PG545 and increased in SST0001 treated groups (Figure 7). Interestingly, both inhibitors resulted in a marked decrease in mRNA levels of LC3 and Casp3, representing lower autophagic activity in the treated livers, compared to E_0_-CD and E_0_-HFD control groups (Figure 7H,I).

Effect of PG545 on oxidative stress. PG545 caused a significant dose-dependent reduction in serum OS, evident by decreasing lipid peroxide (PD) content from 986 ± 16 nmol/mL in the E_0_ control group to 871 ± 37 nmol/mL (*p* = 0.02) and to 746 ± 22 nmol/mL (*p* < 0.001) in the low and high dose treated groups, respectively, and by decreasing TBARS levels from 27 ± 1 nmol MDA/mL in the E_0_ control group to 20 ± 2 nmol MDA/mL (*p* = 0.01) and to 12 ± 0.6 nmol MDA/mL (*p* < 0.001) in the low and high dose treated groups, respectively (Table 2). 

Effect of heparanase inhibition on liver fibrosis. Picrosirius red and Masson’s Trichrome stainings were applied on liver sections. In specimens of mice livers from PG545 and SST0001 treated groups, normal liver architecture was maintained, compared to markedly distorted liver tissue in the E_0_-HFD control group, where significant steatosis was demonstrated. No significant fibrosis was demonstrated in any one of the samples (Figure 2). Gene expression analysis revealed a marked decrease in FGF-2 levels, HIFI1 and SDC1, in the treatment groups compared to control (Figure 8). 

Effect of heparanase inhibition on heparanase level in the liver. Immunofluorescent staining for heparanase revealed intense staining for the enzyme in livers of normal (wild type) and PG545 treated groups, which was mainly intracellular. In contrast, weak staining was demonstrated in the livers of E_0_ control mice, in which the staining was mainly in the extracellular spaces (Figure 8).

Effect of heparanase inhibition on biochemical markers. Both inhibitors resulted in a significant reduction in serum levels of TC, LDL, HDL, and TG (Figure 4A–D). 

## 4. Discussion

Triglycerides are the main lipid component involved in liver steatosis. HSPGs regulate the clearance of serum triglyceride-rich particles (TRPs) via binding, uptake, and degradation of TRPs in the liver [45]. Internalization of lipid remnants occurs via several pathways, including those transduced by LDL receptor (LDLR), LDLR-related protein (LRP), CD36, and HSPGs [46,47]. HSPGs also facilitate the internalization of apo-E enriched remnant lipoproteins by an LDLR independent pathway [48].

In accordance with our previous study, where we demonstrated that PG545 significantly abolished fat accumulation in the liver and decreased plasma levels of TC and TG [38], SST0001 in the current study also significantly attenuated fat accumulation in the liver, albeit to a lesser extent than PG545. As heparanase is implicated in the modulation of HSPG chains, inhibiting heparanase activity altered the uptake and processing of lipids by the liver, and resulted in attenuation of hepatosteatosis development. Studies in liver tissue revealed that SST0001 significantly decreased liver cholesterol and TG content, and decreased cholesterol and TG uptake by MPMs, an additional mechanism by which heparanase inhibition attenuates liver steatosis. In addition, heparanase inhibition resulted in marked decreased lipid uptake parameters in the liver, including DGAT1, ApoB, MTTP, LDLR, LRP, and CD36, as well as lipid synthesis HMGCoA reductase. Similar results were obtained in studies on MPMs derived from animals, where heparanase inhibition significantly decreased cholesterol and TG content. Heparanase inhibition also decreased IL6 and TNF-α, as well as beta-oxidation of cholesterol, evident by lower CPT1a and PPAR-Ɣ levels. 

Both PG545 and SST0001 significantly lowered serum levels of TC, HDL, and LDL-C. TG levels were significantly lowered by PG545 but not SST0001. Both inhibitors significantly lowered TC and TG content in the liver and MPMs. These findings shed light on the results of our former study, in which PG545 significantly lowered serum TC and TG levels in E_0_ mice placed on HFD [38], and in line with the study of Planer et al. [45], where heparanase over-expression resulted in reduced hepatic clearance of postprandial lipoproteins and higher levels of fasting and postprandial serum TG. Lowered lipid uptake by MPMs could be the basis of the lower atherosclerosis development demonstrated in our former study [38] and contributes to the attenuation of liver steatosis observed in the current study. 

In summary, our data demonstrate that heparanase inhibition resulted in both decreased lipid uptake by liver cells and decreased lipid oxidation, leading to a significant reduction in the development of liver steatosis.

The effect of Heparanase inhibition on liver fibrosis was assessed by using Picrosirius red and Masson Trichrome stainings and measuring the pro-fibrotic cytokines FGF-2 and protein kinase B (AkT) in liver homogenates. Heparanase inhibitors restored liver tissue and maintained normal liver architecture in comparison to the control group, in which marked hepatic steatosis had developed. No significant fibrosis was noted in any of the studied groups. Gene expression studies revealed significantly decreased FGF-2 expression in the PG545 and to a lesser extent in the SST0001 treated mice compared to control, as well as significantly lower levels of SDC1, representing lower ECM deposition. No fibrotic response was demonstrated in the liver. Similarly, both doses of PG545 significantly reduced AKT compared to control. AKT is implicated in HSC activation, cell proliferation, and collagen synthesis, all promoting the progression of hepatic fibrosis [49]. AKT is also believed to be implicated in immune cell activation by regulating key inflammatory cytokines, including TNF-α, IL-6, and IL-8 [50]. The decreased levels of AKT obtained in our study may contribute to the favorable effect of heparanase inhibition towards preventing liver fibrosis, as lower AKT levels result in reduced expression of the inflammatory cytokines, which upon prolonged activation may induce advanced liver fibrosis.

The anti-inflammatory effect of heparanase inhibition was assessed by qPCR, measuring the expression of pro-inflammatory cytokines in liver homogenates. TNF-α can trigger multiple signaling pathways involved in inflammation, proliferation, and apoptosis. Thus, it is widely accepted that TNF-α is implicated in chronic liver inflammation that leads to liver fibrosis. Support for this notion is derived from the observation that the inflammatory phase is perpetuated by TNF-α production, which results in the activation of resident HSCs into fibrogenic myofibroblasts. In our study, PG545 significantly reduced the level of TNF-α, and both heparanase inhibitors decreased Casp3 and LC3 levels, effects that are expected to suppress the inflammatory and the pro-fibrotic responses and result in decreased ECM deposition, leading to prevention of liver fibrosis. The expression of heparanase in liver fibrosis was investigated only in one study, which showed increased expression of the heparanase gene in the zebrafish model of HCV-related liver fibrosis [51]. Our findings concerning the effects of heparanase inhibition on pro-inflammatory cytokines are in line with those described by Zhao et al. [51]. Moreover, in a mouse model of CCl_4_-induced liver fibrosis, Secchi et al. demonstrated co-localization of TNF-α with heparanase in the early stage of liver injury, suggesting a key role of heparanase in sustaining inflammation and fibrosis [52].

Similarly, IL-6 is produced by various cell types and acts on many types of target cells in the immune and hematopoietic systems. In the present study, PG545 caused a significant reduction in IL-6, TNF-α, and TLR4 levels in the liver. Lowering these pro-inflammatory cytokines is expected to reduce inflammation, as IL-6 induces the synthesis of acute-phase proteins in the liver (such as IL-1, TNF-α, and phospholipase A2) and is involved in the pathogenesis of fibrosis induction. In their study, Choi et al. demonstrated increased collagen and ECM synthesis by HSCs in rats following intraperitoneal injection of human recombinant IL-6, besides excess hepatic inflammation and fibrosis [53]. In line with these findings, we demonstrated that PG545 significantly decreased IL-6 and LC3 levels in liver homogenates, reflecting an additional favorable effect of heparanase inhibition towards preventing the development of liver fibrosis. Both inhibitors significantly decreased levels of TLR4 and CD163 and increased the levels of the anti-inflammatory cytokine IL4.

Heparanase inhibition and serum oxidative stress. OS is implicated in carcinogenesis, inflammation, and atherosclerosis. Both OS and inflammation are crucial in the initiation and progression of liver pathologies [54]. OS causes hepatic damage by provoking alteration of biological molecules such as DNA, proteins, and lipids, and, notably, modulating biological pathways associated with gene transcription, protein expression, cell apoptosis, and HSC activation [55]. In our former studies, we demonstrated that PG545 significantly reduced both OS and atherosclerosis progression [37,38]. Thus, heparanase inhibition prevents the destructive effect of OS in the liver and contributes to decreasing liver steatosis and fibrosis.

Heparanase inhibition significantly decreased LC3 level, a marker of active autophagy, which reflects the death of HSCs, besides significant reduction in LC3 gene expression. To the best of our knowledge, this is the first description in the literature regarding the effect of heparanase inhibition on liver fibrosis, as searching “MEDLINE” revealed no matching articles.

## 5. Conclusions

Liver steatosis and fibrosis develop in consequence of accelerated inflammatory and oxidative stress and occur following activation of several pro-inflammatory and pro-fibrotic cytokines. In our study, heparanase inhibition caused a significant reduction in the levels of key pro-inflammatory and pro-fibrotic cytokines, significantly reduced lipid uptake by the liver and MPMs, and lowered the levels of serum OS and lipid oxidation.

In summary, heparanase inhibitors apparently have the potential to treat cases of liver steatosis and fibrosis as well as atherosclerosis and are worth being further investigated as a new therapeutic modality in clinical trials.

## 6. Limitations

Mouse model used: To study the effect of PG545 on liver fibrosis, the mouse model of CCl_4_, induced liver fibrosis, should be utilized, as animals in this model rapidly develop severe liver fibrosis upon exposure to CCl_4_, whereas E_0_ mice are not a fibrosis model;Sample size: We used a small number of animals in each treatment group. Increasing the number of animals in the study groups would absolutely strengthen our results. However, it is noteworthy to emphasize the consistent effect of heparanase inhibition in all the animals in the same group.

## Figures and Tables

**Figure 1 jcm-11-01672-f001:**
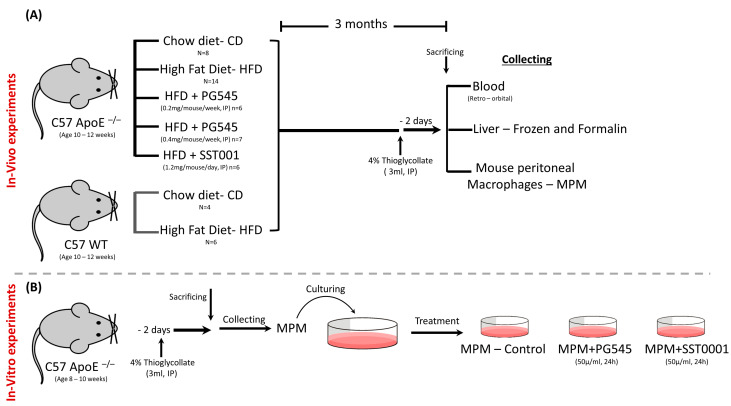
Study methodology. (**A**) In vivo experiments. C57 ApoE^−/−^ mice were placed on CD or HFD; the latter were treated with PG545 in two doses, SST0001 or NS. C57 WT mice were placed on either CD or HFD. Mice were then sacrificed, and blood, livers, and MPMs were collected for analysis. (**B**) In vitro experiments. MPMs were collected from C57 ApoE^−/−^ mice, treated with PG545 or SST0001 and tested for lipid content and beta-oxidation of lipids.

**Figure 2 jcm-11-01672-f002:**
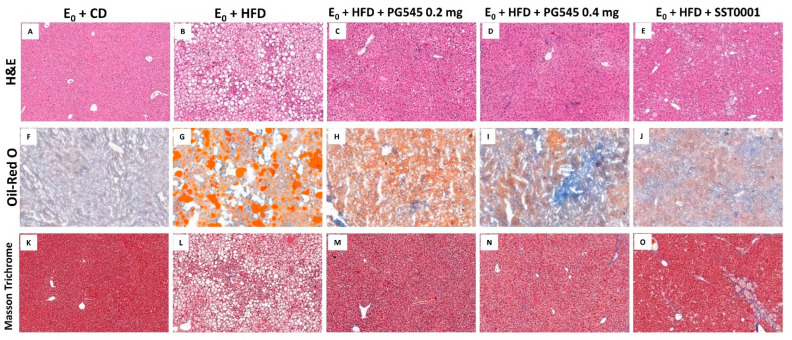
Effect of heparanase inhibition on liver steatosis. H&E, Oil-Red-O, and Masson’s Trichrome stainings of liver sections from E_0_ mice on CD (**A**–**C**), E_0_ mice on HFD (**D**–**F**), E_0_ mice on HFD and PG545 low dose (**G**–**I**), E_0_ mice on HFD and PG545 high dose (**J**–**L**), and E_0_ mice on HFD and SST0001 (**M**–**O**).

**Figure 3 jcm-11-01672-f003:**
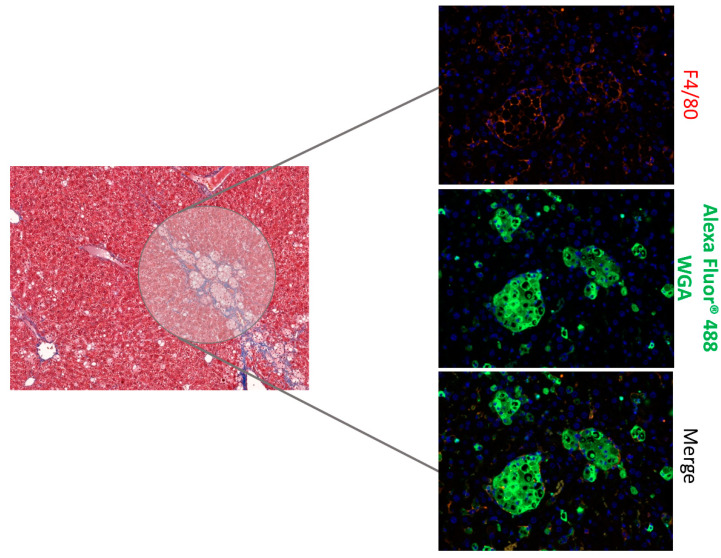
Effect of SST0001 on liver histology. Immunoflourescent staining of liver sections with macrophages marker (F4/80 antibody) and wheat germ agglutinin (WGA, a carbohydrate-binding lectin) showing granuloma-like structures.

**Figure 4 jcm-11-01672-f004:**
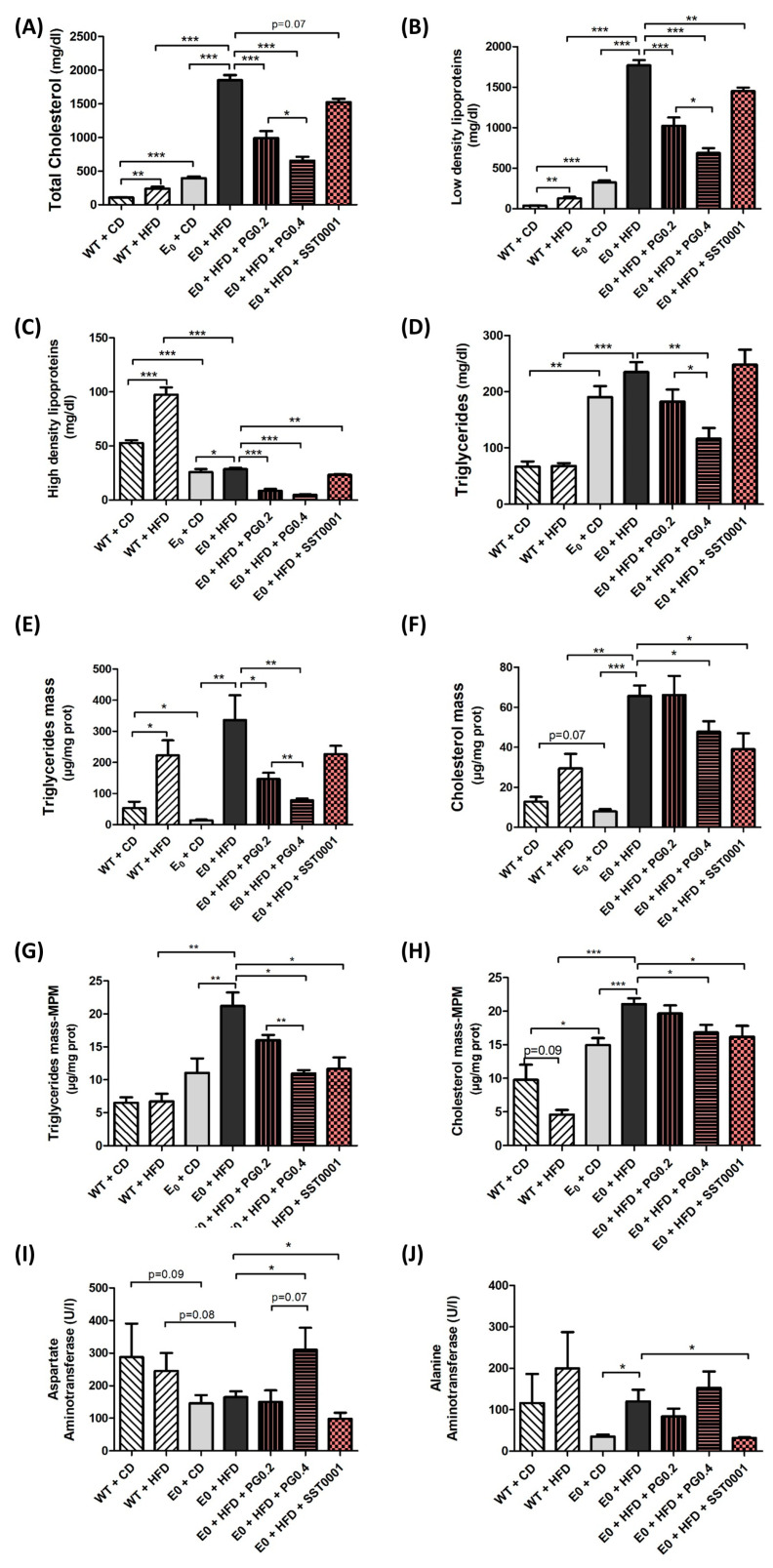
Effect of heparanase inhibitors on lipid uptake in macrophages. Cholesterol, LDL-C, HDL-C, and TG serum levels (**A**–**D**), TG and cholesterol content in the liver (**E**,**F**), TG and cholesterol in MPMs in vivo (**G**,**H**), and liver enzymes in serum (**I**,**J**). * *p* < 0.05, ** *p* < 0.01 and *** *p* < 0.001.

**Figure 5 jcm-11-01672-f005:**
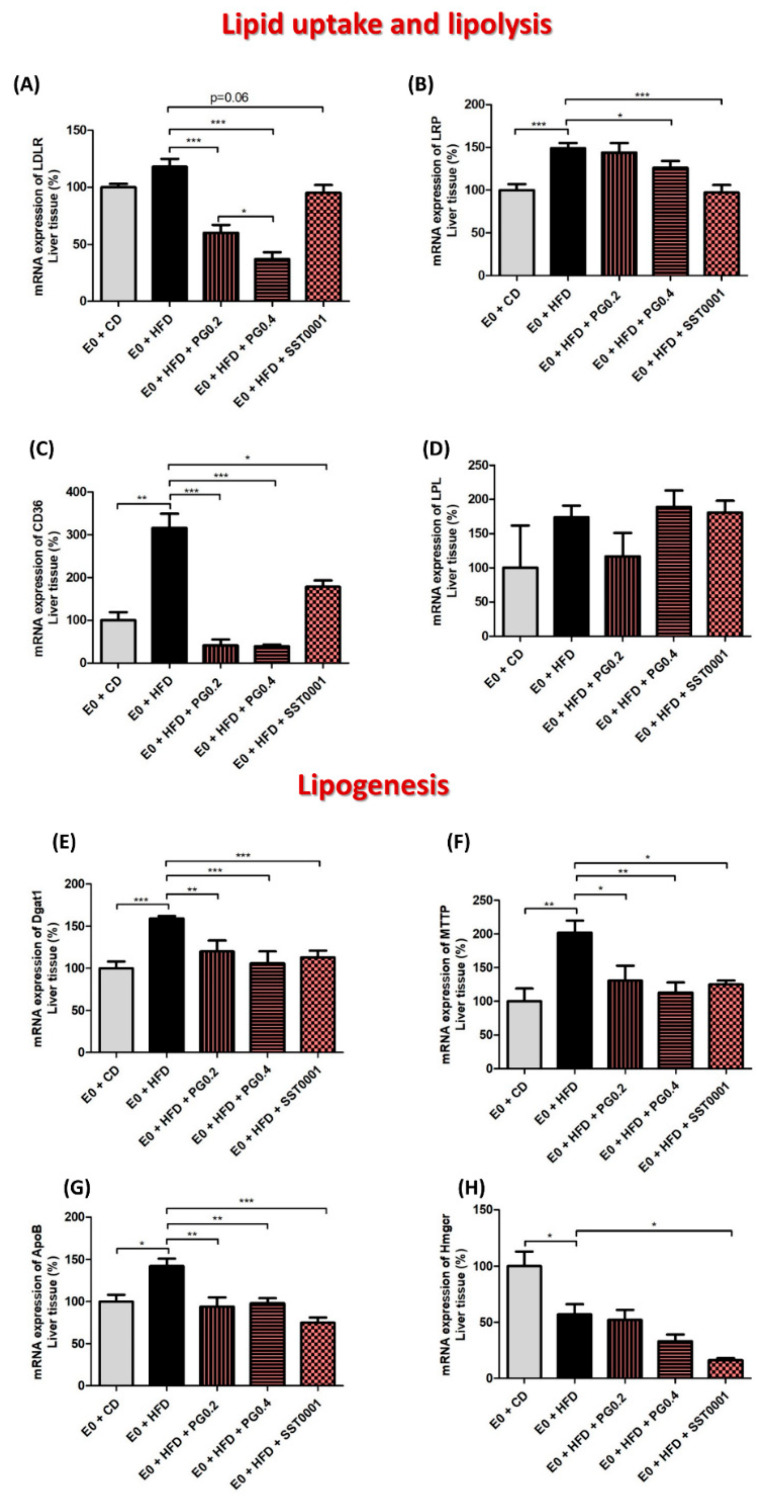
Effect of heparanase inhibitors on lipid metabolism in the liver. mRNA expression of lipid metabolism pathway markers in liver (**A**–**H**). * *p* < 0.05, ** *p* < 0.01, *** *p* < 0.001.

**Figure 6 jcm-11-01672-f006:**
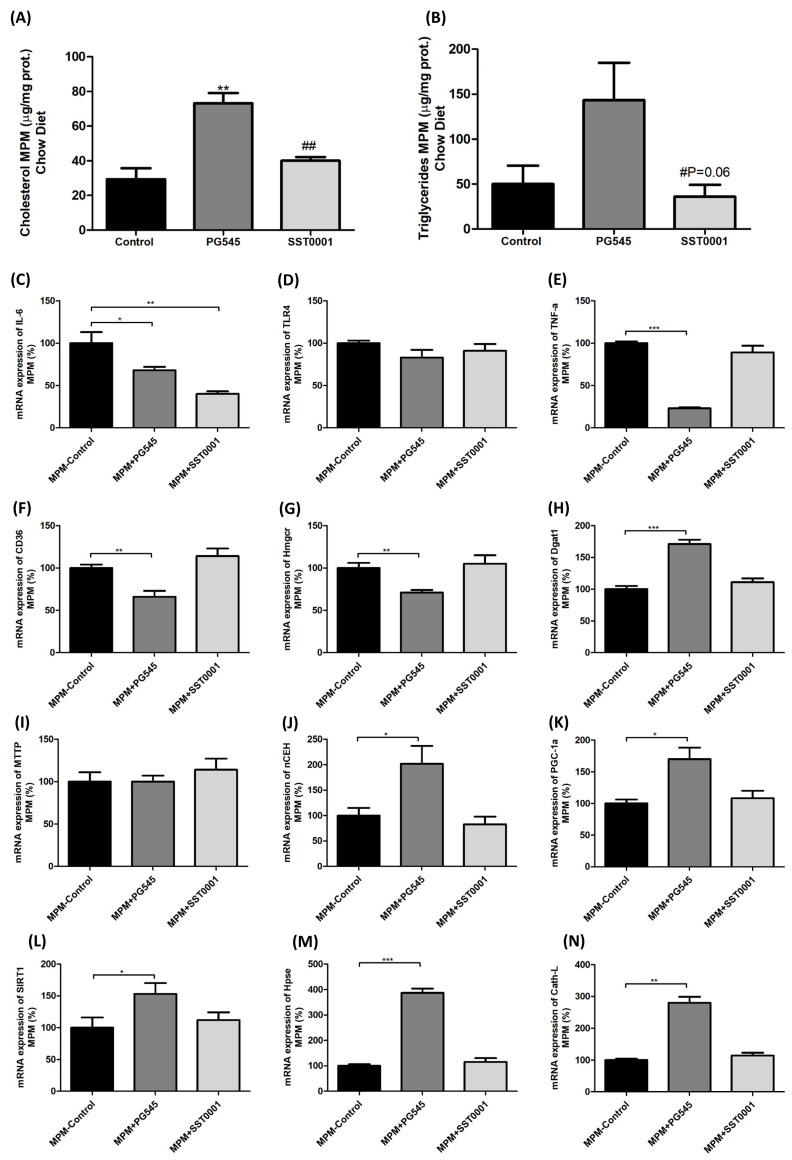
Effect of heparanase inhibitors on macrophage lipid uptake and inflammation (in vitro). TC and TG content (**A**,**B**), mRNA expression of inflammation markers (**C**–**E**) and lipid uptake and metabolism markers (**F**–**N**). (*) as compared to Control. (#) as compared to PG545. * *p* < 0.05, ** *p* < 0.01, *** *p* < 0.001. # *p* < 0.05, ## *p* < 0.01.

**Figure 7 jcm-11-01672-f007:**
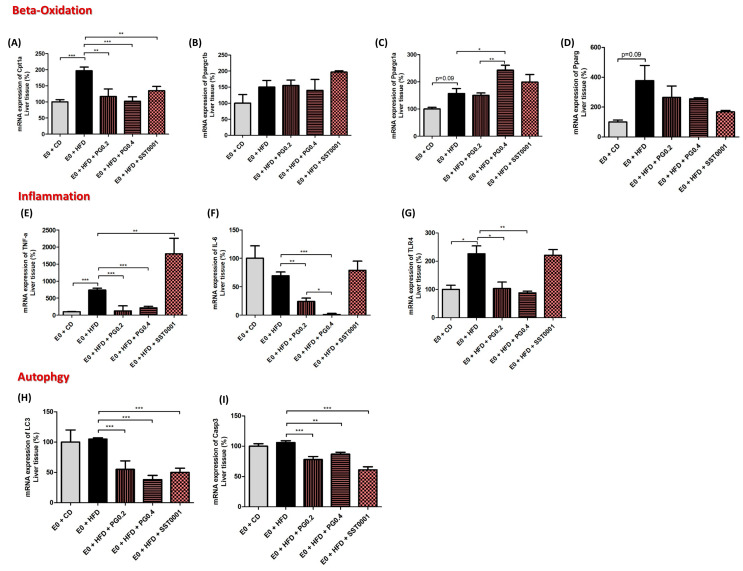
Effect of heparanase inhibition on lipid oxidation (**A**–**D**), inflammatory stress (**E**–**G**) and autophagy (**H**,**I**) in the liver. * *p* < 0.05, ** *p* < 0.01, *** *p* < 0.001.

**Figure 8 jcm-11-01672-f008:**
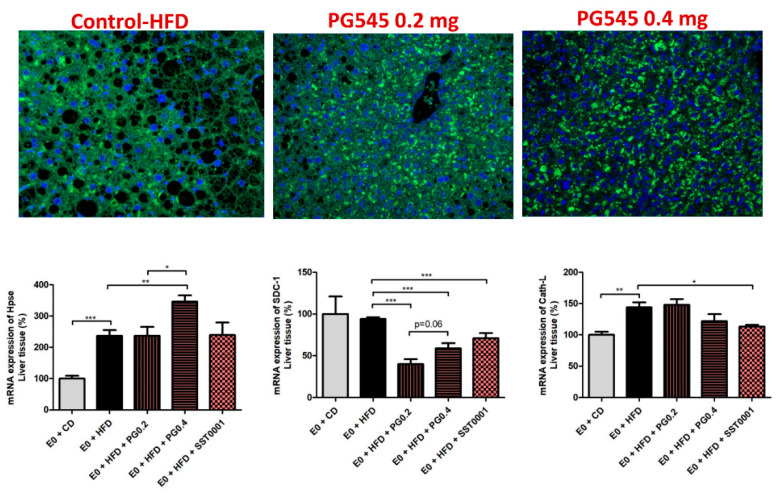
Effect of heparanase inhibtion on liver fibrosis. * *p* < 0.05, ** *p* < 0.01, *** *p* < 0.001.

**Table 1 jcm-11-01672-t001:** Sequence of forward and reverse primers used in qPCR experiments.

		Forward	Reverse	Amplicon Length
1	LDLR	TCGACTCACGGGTTCAGATG	ACCAGTTCACCCCTCTAGGC	104
2	LRP1	CTCCCACCGCTATGTGATCC	TCGCTGCCCACATACTTGTT	132
3	CD36	CATTTGCAGGTCTATCTACG	CAATGTCTAGCACACCATAAG	182
4	LPL	GGATGGACGGTAACGGGAAT	ATAATGTTGCTGGGCCCGAT	123
5	DGAT1	TCATACTCCATCATGTTCCTC	GAAGTAATAGAGATCTCGGTAGG	174
6	MTTP	TCCTGGACTTTTTGGATTTC	TTGAACTTACTAAGGAGGGC	124
7	ApoB	CTCCTACAAGAATAAGTATGGG	GAAGCGACTGTTGATCTTAG	85
8	HMGCoA reductase	GATAGCTGATCCTTCTCCTC	ATGCTGATCATCTTGGAGAG	135
9	IL-6	AAGAAATGATGGATGCTACC	GAGTTTCTGTATCTCTCTGAAG	164
10	TLR4	TCCCTGCATAGAGGTAGTTCC	TCCAGCCACTGAAGTTCTGA	171
11	TNF-a	CTATGTCTCAGCCTCTTCTC	CATTTGGGAACTTCTCATCC	109
12	nCEH	CTAGTGCAAAGATCAGCTAC	ATTTGCTCCGGAAAATAGAC	118
13	SIRT1	AAACAGTGAGAAAATGCTGG	GGTATTGATTACCCTCAAGC	75
14	HPSE	CCAAGTGCTCGGGGTTAGAC	AGAAACTGTTGGGCTCATTGC	159
15	Cath L	CCCTATGAAGCGAAGGACGG	CTGGAGAGACGGATGGCTTG	165
16	Cpt1a	GGGAGGAATACATCTACCTG	GAAGACGAATAGGTTTGAGTTC	183
17	pPARgc1b	AAGAACTTCAGACGTGAGAG	TCAAAGCGCTTCTTTAGTTC	161
18	pPARgc1a	TCCTCTTCAAGATCCTGTTAC	CACATACAAGGGAGAATTGC	78
19	pPARg	AAAGACAACGGACAAATCAC	GGGATATTTTTGGCATACTCTG	195
20	LC3	GAACCGCAGACGCATCTCT	TGATCACCGGGATCTTACTGG	171
21	Casp3	GAGTCCACTGACTTGCTCCC	AGCTTGGAACGGTACGCTAA	117
22	Rpl13a	AAGCAGGTACTTCTGGGCCG	GGGGTTGGTATTCATCCGCT	126

**Table 2 jcm-11-01672-t002:** Biochemical parameters in sera of mice groups. PG545 caused dose-dependent decrease in levels of TC, TG, LDL, HDL, and oxidative stress parameters (marked in red). Levels of glucose, creatinine, and liver enzymes were not significantly affected by the inhibitor. Results are expressed as mean ± SEM.

	Control (*n* = 5)	PG545 (mg/Mouse)		*p*-Value		
		0.2 (*n* = 5)	0.4 (*n* = 5)	Control vs. 0.2	Control vs. 0.4	0.2 vs. 0.4
TC (mg/dL)	652 ± 21	397 ± 29	274 ± 13	0.0001	*p* < 0.0001	0.0055
TG (mg/dL)	230 ± 23	189 ± 24	136 ± 20	0.2677	0.0169	0.1326
LDL (mg/dL)	542 ± 19	339 ± 22	237 ± 10	0.0003	*p* < 0.0001	0.0035
HDL (mg/dL)	53 ± 0.8	19 ± 3.6	10 ± 0.8	*p* < 0.0001	*p* < 0.0001	0.0337
PD (nmol/mL)	985 ± 15	870 ± 36	746 ± 21	0.0214	* p * < 0.0001	0.0196
TBARS (nmol MDA/mL)	27.3 ± 1.0	19.5 ± 2.0	12.3 ± 0.6	0.0101	* p * < 0.0001	0.0103
PON1 activity (U/mL)	532 ± 15	326 ± 28	226 ± 16	0.0002	* p * < 0.0001	0.0156
Glucose	166 ± 10	178 ± 13	159 ± 11	NS ^†^	NS ^†^	NS ^†^
Creatinine	0.41 ± 0.01	0.41 ± 0.01	0.40± 0.01	NS ^†^	NS ^†^	NS ^†^
Blood Urea Nitrogen (mg/dL)	26.0 ± 0.7	24.2 ± 1.4	30.0 ± 2.2	NS ^†^	NS ^†^	NS ^†^
ALT (U/L)	161 ± 13	66 ± 7	108 ± 24	0.0003	0.0900	0.1379
AST (U/L)	149 ± 7	116 ± 12	223 ± 31	0.0540	0.0525	0.0135
ALP (U/L)	54 ± 4	57 ± 7	66 ± 4	0.7595	0.0832	0.3106

^†^ NS—non-significant.

## Data Availability

Data supporting reported results can be found, including links to publicly archived datasets analyzed or generated during the study with the corresponding author.

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
