# Peer review of "Heparanase Inhibition Prevents Liver Steatosis in E0 Mice"

_jcm, 2022, doi:10.3390/jcm11061672_

Round 1

Reviewer 1 Report

The work presented reports the effect of heparanase inhibition on the development of liver steatosis and fibrosis in E0 mice placed on high-fat diet, focusing on possible mechanisms by which heparanase inhibition exerts these effects. It is an interesting manuscript, promising interest to the scientific community so I recommend some concerns that need to be further addressed.
1. Abstract is written a little complicated, please seriously revise and make it more outstanding rather than just copying from the results chapter directly. 
2. Why the animal cohorts (number) were different between each group? To have a statistic, the number of mice MUST be to be same
3. Text in Figure 1 needs a larger size, especially Fig 1A. Also, Fig. 4 is hard to read
4. Immunoblotting should be further applied to verify their significant findings since the current results did not support their conclusions as to the reduction in the levels of key pro-inflammatory and pro-fibrotic cytokines...

Author Response

10-MAR-2022

Please find attached our response to reviewer’s comments, besides the updated manuscript uploaded in the journal site.

Response to reviewer 1:

  1. Abstract section has been changed in accordance with the reviewer’s comments. Please see new version submitted to the journal site.
  2. Number of animals and statistics.
    1. There were small differences in the number of the animals in each group (most groups were with 6-8 animals). Indeed, we had a limited number of animals, and tried to choose as much as possible for each group, through utilizing all the available animals. There were no fatalities in the course of the study.
    2. While processing the statistics, we applied one-way ANOVA for non-repeated measurements, which does not require equal number in each experimental group.
  3. Figures were submitted in new version, hope to be easier and more clear to read.
  4. For technical difficulties, immunoblotting studies were performed only in the subgroups of E0 mice which were on either Chow diet or HFD with the PG5454, but not on the SST0001 mice groups. Western blotting analysis showed that PG545 significantly reduced the level of TNF-α in the liver homogenates, p=0.02 compared to control. Low dose of PG545 caused no significant alteration in the level of TNF-α, p=0.43, (Fig.2 A&F). IL-6 levels were significantly decreased in the PG545 treated groups, p=0.002 and p<0.001 in the low and high dose treated groups, respectively, (Fig.2 B&G). Immunoreactive levels of FGF-2 and Akt were measured in liver homogenates. PG545 caused a significant dose-dependent elevation in FGF-2 abundance in the PG545 treated groups, as compared to control (p=0.003 and p<0.001) in the low and high dose treated groups, respectively (Fig.2 C&H). In contrast, PG545 caused a significant dose-dependent decrease in Akt abundance in the PG545 treated groups compared to control, p<0.001 in both treated groups compared to the control group (Fig.2 D&I). These data were not included in the MS due to lack of results in the SST0001 treated mice group. Attached please find figure showing the results of western blotting studies in these groups that was uploaded as supplementary figure:

Inflammatory reactions in the liver. The effect of PG545 on the inflammatory process was assessed by measuring the proinflammatory cytokines level in the liver tissue. Overall, PG545 decreased the levels of TNF-α and IL-6. Only the high dose of PG545 significantly reduced the level of TNF-α in the liver homogenates, p=0.02. The low dose of PG545 caused no significant alteration in the level of TNF-α, p=0.43, (Fig.2 A&F). IL-6 levels were significantly decreased in the PG545 treated groups, p=0.002 and p<0.001 in the low and high dose treated groups, respectively, (Fig.2 B&G).

Reviewer 2 Report

In the study “Heparanase Inhibitor Prevents Liver Steatosis in Eo Mice” Kinaneh et al. propose two types of inhibitors against liver steatosis and fibrosis in Eo mice fed with HFD focusing on mechanisms based on fat uptake and fat metabolic pathways as well as inflammation (KC activation pathway), which finally would trigger to fibrosis.

  • They perform the experiments in C57 male mice, however size of the groups seems unequal, Could the authors give a potential explanation of this? Is there any mortality associated to Eo mice fed with HFD?
  • Quality of the images should be definitely improved. In figure 2, it is impossible to distinguish anything in the pictures due to the quality and pixelation of the images. Also the headings in the image have poor quality.
  • Poor quality of the graphs in the figure 4. It is impossible to read which information is contained on them.
  • Did the authors checked if the inhibitors has any effect in the Eo mice placed with chow diet? Normal levels of WT are reached?
  • Figure 6. Labelling of the groups in figure 6A and 6B is different from the C-N figure 6.
  • Figure 7. Graphs have different sizes. Also the quality needs to be improved. 
  • Did de authors checked if the inhibitors are also playing a role in the oxidative stress induced by lipid metabolism that would also triggered to hepatocyte cell death and induce higher levels of inflammation? 

Author Response

10-MAR-2022

Please find attached our response to reviewer’s comments, besides the updated manuscript uploaded in the journal site.

Response to reviewer 2:

  1. Number of animals. There were small differences in the number of the animals in each group (most groups were with 6-8 animals). Indeed, we had a limited number of animals, and tried to choose as much as possible for each group, through utilizing all the available animals. There were no fatalities in the course of the study.
  2. Figures were submitted in new version, hope to be easier and more clear to read.
  3. We did not check the inhibitors in E0 mice placed on Chow diet.
  4. Figures were submitted in new version, hope to be easier and more clear to read.
  5. For technical limitations, Oxidative stress was measured only in the subgroups of E0 mice which were on either Chow diet or HFD with the PG545, but not in the SST0001 mice group. PG545 caused a significant dose-dependent reduction in serum OS, evident by decreasing lipid peroxide (PD) content from 986±16 nmol/ml in the E0 control group to 871±37 nmol/ml (p=0.02) and to 746±22 nmol/ml (p<0.001) in the low and high dose treated groups, respectively, and by decreasing TBARS levels from 27±1 nmol MDA/ml in the E0 control group to 20±2 nmol MDA/ml (p=0.01) and to 12±0.6 nmol MDA/ml (p<0.001) in the low and high dose treated groups, respectively (Table 1s). Attached please find results of OS measurements in the following table which was included as supplementary table, marked with red color:

Control (n=5)

PG545 (mg/mouse )

p-value

0.2 (n=5)

0.4 (n=5)

Control vs 0.2

Control vs 0.4

0.2 vs 0.4

TC (mg/dL)

652 ± 21

397 ± 29

274 ± 13

0.0001

P<0.0001

0.0055

TG (mg/dL)

230 ± 23

189 ± 24

136 ± 20

0.2677

0.0169

0.1326

LDL (mg/dL)

542 ± 19

339 ± 22

237 ± 10

0.0003

P<0.0001

0.0035

HDL (mg/dL)

53 ± 0.8

19 ± 3.6

10 ± 0.8

P<0.0001

P<0.0001

0.0337

PD (nmol/ml)

985 ± 15

870 ± 36

746 ± 21

0.0214

P<0.0001

0.0196

TBARS (nmol MDA/ml)

27.3 ± 1.0

19.5 ± 2.0

12.3 ± 0.6

0.0101

P<0.0001

0.0103

PON1 activity (U/ml)

532 ± 15

326 ± 28

226 ± 16

0.0002

P<0.0001

0.0156

Glucose

166 ± 10

178 ± 13

159 ± 11

NS

NS

NS

Creatinine

0.41±0.01

0.41±0.01

0.40± 0.01

NS

NS

NS

Blood Urea Nitrogen (mg/dL)

26.0±0.7

24.2±1.4

30.0±2.2

NS

NS

NS

ALT (U/l)

161 ± 13

66 ± 7

108 ± 24

0.0003

0.0900

0.1379

AST (U/l)

149 ± 7

116 ± 12

223 ± 31

0.0540

0.0525

0.0135

ALP (U/l)

54 ± 4

57 ± 7

66 ± 4

0.7595

0.0832

0.3106

Table 1s: biochemical parameters in sera of mice groups. PG545 caused dose dependent decrease in levels of TC, TG, LDL, HDL and oxidative stress parameters (marked in red). Levels of Glucose, creatinine and liver enzymes were not significantly affected by the inhibitor. Results are expressed as mean±SEM.

NS - non-significant

Round 2

Reviewer 1 Report

Thanks for revising the manuscript. I have no additional remarks. Please mention those limitations in the conclusion as my minor comment.